# BEAST: Leveraging Contrastive Learning and Unsupervised Sentence Embeddings for Improved Drug Abuse Detection

## Abstract

Prescription drug abuse is a growing public health crisis worldwide. In the digital age, social media platforms offer a unique opportunity to monitor drug abuse trends in real-time. However, traditional machine learning models struggle with the informal language, sarcasm, and figurative speech used on social media. This paper proposes BEAST, a novel approach that leverages contrastive learning to improve the detection of drug abuse references hidden within figurative language. Additionally, the integration of SimCSE and Target-Based Generating Strategy further enhances the model's performance by generating superior representations from both labeled and unlabeled data. We test our model on three datasets, and the experimental results demonstrate the superiority of BEAST over the baseline in accurately identifying drug-related references hidden within figurative language on social media. Our work paves the way for more effective public health interventions in this increasingly digital era.

## Keywords

Drug Abuse Detection, Evaluation, Sentence Embeddings, Computational Social Science

**ACM Reference Format:**
Anonymous WWW Submission. 2025. BEAST: Leveraging Contrastive Learning and Unsupervised Sentence Embeddings for Improved Drug Abuse Detection. In *Proceedings of The Web Conference (WebConf '25).* ACM, New York, NY, USA, 7 pages. https://doi.org/XXXXXXX.XXXXXXX

## 1 Introduction

The misuse and abuse of prescription drugs have become a pervasive public health crisis worldwide [4]. Opioids, stimulants, and other prescription medications are increasingly diverted from their intended therapeutic use, leading to addiction, overdose, and even death. According to statistics from the CDC, the number of deaths resulting from drug overdose is growing annually and by 2020, it had reached over 90,000 [3]. While traditional methods for monitoring this issue such as emergency room data and pharmacy sales records are often used as indicators, they may hinder effective intervention by providing outdated or lagging data. This has placed substantial strain on healthcare systems in many countries, prompting a growing demand for machine learning (ML) approaches to understanding how, when, and why prescription drugs are misused.

In today's digital landscape, social media platforms have emerged as a unique and dynamic source of real-time data on human behavior and health [14]. These platforms offer a remarkable opportunity to gain insights into individual and public sentiment, current trends, and potential risks. Posts on social media networks act as a concise and fast-paced currency of communication, which can be a valuable tool for the monitoring of prescription drug abuse by public health researchers and authorities. However, the heterogeneity of user data on social media presents significant challenges to traditional machine learning (ML) models [18]. The informal nature of online communication often deviates from structured language patterns and includes linguistic intricacies such as sarcasm, slang, and metaphors [19]. This is particularly problematic when attempting to identify references to prescription drug abuse, as users frequently resort to euphemisms and metaphors to conceal their actions. Nevertheless, the wealth of data available from social media platforms, coupled with advanced natural language processing (NLP) techniques, holds the potential to substantially improve the monitoring and early detection of prescription drug abuse trends.

Early NLP-driven solutions, including simpler rule-based or keyword-based approaches, often struggle with the dynamic and informal nature of social media language [6]. While contemporary ML methods demonstrate promise in analyzing social media text, they often struggle with detecting references hidden within figurative language. Pre-trained models such as Bidirectional Encoder Representations from Transformers (BERT) [5] are often effective in identifying explicit references, but their performance declines when encountering metaphorical language [12]. This limitation significantly hinders the effectiveness of such models for public health monitoring.

Contrastive learning [10] can be particularly effective in tasks like drug abuse detection, where figurative language may obscure intent, signified by methods such as CATE [11], a contrastive pre-trained model for metaphor detection, and CLINE [22], which uses semantic negative examples for better sensitivity to language nuances. SimCSE [7] further advances contrastive learning with unsupervised and supervised techniques, improving sentence embeddings and proving effective for tasks like sentiment analysis, relevant to drug abuse detection from social media posts.

Driven by the advancements in contrastive learning and to address the gaps in public health monitoring, we propose BEAST, a model that implements unsupervised contrastive learning to enhance drug abuse detection on social media. Initially, BERT is employed to obtain semantic representations from the text. We then self-train the model using a Target-Based Generating Strategy (TGS) to obtain an expanded dataset with candidate instances of target words. Finally, we use SimCSE to enhance sentence embeddings by constructing positive and negative pairs using synonym and antonym replacement. Contrastive learning also improves the model's ability to differentiate between similar and dissimilar examples, which helps the model better capture the semantic nuances and distinguish between

metaphorical and literal meanings. This enables the model to judge sentence similarity more accurately and improve its performance on social media posts. Our findings underscore the importance of capturing nuanced contexts and semantic shifts, thereby enabling health authorities and researchers to glean valuable insights from digital discussions around prescription drug misuse.

## 2 Related Work

### 2.1 Social Media Analysis for Drug Abuse Detection

The use of social media platforms, particularly Twitter, for monitoring and detecting prescription medication abuse, has gained significant attention in recent years. In a study analyzing 2,100 tweets related to prescription opioids, [20] categorized the tweets into three groups: abuse (use for recreational purposes), non-abuse (use for medical purposes), or unclassifiable. Additionally, each tweet was evaluated based on its connotation—whether it promoted psychoactive or analgesic use (positive), described an adverse event (negative), or was not characterizable. The findings indicated that discussions of abuse were prevalent, with over 66% of the tweets reflecting a positive connotation toward opioid use. Various NLP and machine learning techniques have been applied to analyze social media data and detect patterns indicative of drug abuse. Early works, such as by [17], explored the application of decision-tree-based classifiers and N-gram models to achieve high accuracy in detecting drug-related posts on Twitter. However, their research highlighted that synonym expansion and drug-slang lexicons, when used as single features, did not perform as effectively in improving model performance. Their study laid the foundation for further exploration of social media platforms as viable tools for monitoring drug abuse. Further [16] addressed the challenge of noise in social media data, which can hinder the performance of traditional keyword-based monitoring systems. They evaluated various supervised classification methods, including Support Vector Machines (SVM), Random Forests (RF), and Convolutional Neural Networks (CNN). They utilized drug names as keywords for tweet collection and manually labeled them to identify instances of drug abuse. These labeled data were then used to train supervised models, which proved effective in distinguishing between drug-related and non-abuse-related content. Additionally, they developed a propagation network based on relationships in tweets to monitor the spread of drug-related information. Recent advances have focused on improving the precision of models in distinguishing between literal and figurative mentions of drug-related terms. [2] proposed a pipeline combining pre-trained contextual word representations with sentiment distributions, enabling a more accurate differentiation between figurative and literal usage of drug names.

### 2.2 NLP Advancements in Contrastive Learning

Contrastive learning has emerged as a powerful technique for improving the quality of representation learning. Recent advancements have shown the potential of contrastive learning to enhance the detection of subtle differences in meaning, which is critical for tasks like drug abuse detection where figurative language or euphemisms may obscure the true intent of social media posts. [11] introduced the ContrAstive Pre-trained Model (CATE), a framework designed for metaphor detection using semi-supervised learning.

By first pretraining a model to obtain contextual embeddings and then applying a contrastive objective to distinguish between literal and metaphorical senses of words, CATE achieved state-of-the-art results in metaphor detection. Similarly [22] proposed CLINE, a contrastive learning approach that incorporates semantic negative examples. By replacing keywords with antonyms or synonyms and using adversarial examples, CLINE helps models become more sensitive to subtle semantic differences. This approach is especially relevant for the task of drug abuse detection, where slight changes in language can indicate entirely different meanings. Another recent work is SimCSE[7] which leverages unsupervised and supervised contrastive learning techniques to create better sentence embeddings. In the unsupervised version, different dropout mechanisms create multiple embeddings of the same sentence, which are treated as positive pairs, while other sentences in the batch are treated as negative examples. This unsupervised framework has proven effective for tasks like sentiment analysis, which closely aligns with detecting drug abuse sentiment from social media posts. The ability of SimCSE to enhance sentence-level representations without additional labeled data makes it well-suited for large-scale, noisy datasets like those derived from social media.

## 3 Dataset

Our study uses two categories of data: HMC (Health Mention Classification) and Drug Abuse. HMC identifies whether social media posts with health-related terms discuss actual health issues, and the Drug Abuse dataset detects whether posts mentioning drug names suggest drug abuse.

### 3.1 HMC data

For HMC, we use the dataset introduced by [2]. This dataset extends upon an existing HMC dataset [9] which contains 7,000 tweet IDs related to six diseases. Due to the unavailability of some tweets during data collection, the authors were only able to retrieve 5,497 tweets, additional tweets were gathered using the original six diseases, along with four more disease names. Following this collection and annotation process, 14,061 additional tweets were added, expanding the dataset from 5,497 to 19,558 tweets. After the addition of an extra label 'figurative mention', all the tweets were classified into one of the three classes: 'Health mention', 'Non-Health mention', and 'Figurative mention'. For our experiments, we denote the initial dataset [9] as D1, [2] as D2, and the final extended dataset as D3.

### 3.2 Drug abuse data

For drug abuse detection, we use two different datasets. First, we use an existing dataset from Al et al. [1], which will be denoted as ACMU. The dataset comprised tweets mentioning 20 medications. Alongside drug names, common misspellings were used to gather tweets via the Twitter API[1]. Three trained annotators classified 16,443 tweets into four categories: potential abuse or misuse(A), non-abuse consumption(C), drug mention only(M), and unrelated(U), following detailed annotation guidelines.

Furthermore, we introduce a new annotated drug abuse dataset, ACMU++, with 4,806 tweets. After examining a large number of relevant tweets in the unrelated category, we observed that users

---

[1]https://developer.twitter.com/en/docs/twitter-api

often employ drug names metaphorically. As a result, we decided to reclassify the unrelated category as figurative usage. Consequently, the tweets we collected and labeled fall into four categories: potential abuse or misuse, non-abusive consumption, drug mention only, and figurative usage. The data was collected using the Twitter API and annotated by 4 native English speakers who were given the same annotation guidelines used by [1]. The inter-annotator agreement was high, achieving a Cohen's kappa score of 0.85 on a sample of 300 tweets. The details of the final dataset are present in Table 1.

**Table 1: Drug abuse dataset statistics**

| Label | Original | New | Final |
|---|---|---|---|
| Drug Abuse | 2,636 | 808 | 3,444 |
| Non-Abuse Consumption | 4,589 | 307 | 4,896 |
| Drug Mention | 8,563 | 2,791 | 11,354 |
| Figurative Usage | 655 | 900 | 1,555 |
| **Total** | **16,443** | **4,806** | **21,249** |

## 4 Methodology

### 4.1 Pre-trained Representation Learning

The foundation of our model is the Bidirectional Encoder Representations from Transformers (BERT) architecture [5], a pre-trained language model that achieves state-of-the-art results on various NLP tasks. We finetune a pre-trained instance of BERT to extract semantic representations from the text and encapsulate them into sentence embeddings. Formally, the embeddings $X_{bert} \in \mathbb{R}^{768}$ are obtained from the tweets $T$ as:

$$X_{bert} = BERT(T) \tag{1}$$

### 4.2 Contrastive Learning to Learn Meaningful Representations

To enhance BERT's ability to distinguish between different semantic classes, we incorporate contrastive learning. Contrastive learning aims to learn representations by pulling semantically similar instances closer together while pushing dissimilar instances farther apart. Instead of relying solely on explicit labels, contrastive learning leverages the inherent structure of the data to identify the level of similarity by using positive and negative pairs. Positive pairs are instances that are similar or closely related, while negative pairs are instances that are dissimilar or unrelated. By learning to distinguish between these pairs, the model can learn representations that capture the underlying semantic or structural relationships within the data.

We use the Info Noise Contrastive Estimation (InfoNCE) loss function [15] that encourages the model to produce similar representations for positive pairs and dissimilar representations for negative pairs. The loss $\ell_i$ for sample $i$ is defined as:

$$\ell_i = -\log \frac{e^{\text{sim}(\mathbf{h}_i, \mathbf{h}_i^+)/\tau}}{\sum_{j=1}^{N} e^{\text{sim}(\mathbf{h}_i, \mathbf{h}_j^+)/\tau}} \tag{2}$$

where T is the temperature hyperparameter that determines how much attention the contrast loss pays to the difficult negative samples. When the value of T is greater, the training will favor difficult negative samples less. Instead, it will focus more on the negative samples that are similar to the positive samples and give it a larger gradient in order to separate it from the positive samples to separate it from the positive sample. The *sim* in the function is the cosine similarity, which we use to calculate the similarity of cases in a batch.

We further use the Target-Based Generating Strategy (TGS) as self-training to expand the training data by generating candidate instances for target words. We use contrastive learning and TGS with BERT to obtain our baseline model Contrastive BERT (C-BERT) for enhancing drug abuse detection. C-BERT leverages the power of BERT to capture the semantic meaning of sentences and employs contrastive learning to improve the model's ability to differentiate between similar and dissimilar examples, aiding in metaphor detection.

Despite its strengths, C-BERT exhibits certain limitations for drug abuse detection. The contrastive objective, while beneficial for general-purpose representation learning, may not be as effective in distinguishing between metaphorical and literal meanings. Further, the contrastive learning method is not proficient at capturing similarities between examples of the same class and contrasting them with examples from other classes. Lastly, the model's reliance on extra datasets for candidate generation can be restrictive, limiting its applicability in scenarios where such data is unavailable.

### 4.3 Unsupervised SimCSE

To overcome the limitations of C-BERT, we implement SimCSE [7] to the training process for generating superior sentence embeddings from both labeled and unlabeled data. SimCSE establishes positive and negative pairs in text data by replacing keywords with synonyms and antonyms. This approach enables the model to more accurately judge sentence similarity and improve its performance on social media posts.

A key issue in contrast learning is how to construct positive instance pairs. In NLP, deletion, reordering, and substitution are usually used for data augmentation. However, by integrating unsupervised learning with SimCSE, we only need to use dropout as our data augmentation method to add noise to the text and make positive instance pairs. The dropout mechanism randomly turns off neurons in the network to ensure that the output sentence embedding of the model will have minimal variation if different neurons are turned off. Using the Dropout mask in the BERT model, two different embedding vectors are obtained by forward propagation for each sentence, and the vector pair obtained from the same sentence is used as a positive sample pair. For each vector, the embedding vector generated by the other sentence is selected as a negative sample to train the model. We use cross-entropy to compute the loss, where the inter-sample similarity score is measured for each sample to obtain a score vector with positive samples in the sub-vector having the highest score and vice versa for negative samples.

## 5 Experimental Results

We evaluated the performance of health mention datasets (d1, d2, d3) and drug mention datasets (ACMU, ACMU++) for drug abuse

| Model | HMC_D1 | | | | HMC_D2 | | | | HMC_D3 | | | |
|-------|------|------|------|------|------|------|------|------|------|------|------|------|
| | **P** | **R** | **F1** | **Acc** | **P** | **R** | **F1** | **Acc** | **P** | **R** | **F1** | **Acc** |
| **C-BERT** | 0.91 | 0.90 | 0.89 | 0.92 | 0.73 | 0.71 | 0.71 | 0.84 | 0.83 | 0.82 | 0.81 | 0.88 |
| **BEAST (Ours)** | 0.95 | 0.95 | 0.94 | 0.96 | 0.85 | 0.85 | 0.84 | 0.86 | 0.92 | 0.92 | 0.91 | 0.94 |

**Table 2: Performance of C-BERT and BEAST across HMC_D1, HMC_D2, and HMC_D3 datasets: dropout = 0.1, Poole 0.05, with TGS. P: Precision, R: Recall, F1: Macro F1, Acc: Accuracy**

| Model | D1 binary | | | | D2 binary | | | | D3 binary | | | |
|-------|------|------|------|------|------|------|------|------|------|------|------|------|
| | **P** | **R** | **F1** | **Acc** | **P** | **R** | **F1** | **Acc** | **P** | **R** | **F1** | **Acc** |
| **C-BERT** | 0.96 | 0.95 | 0.95 | 0.95 | 0.84 | 0.83 | 0.83 | 0.92 | 0.92 | 0.91 | 0.91 | 0.91 |
| **BEAST (Ours)** | 0.98 | 0.98 | 0.98 | 0.98 | 0.86 | 0.87 | 0.86 | 0.92 | 0.91 | 0.90 | 0.90 | 0.94 |

**Table 3: Performance of C-BERT and BEAST across D1, D2, and D3 binary classification datasets: dropout = 0.1, Pooler Method = [CLS], Temperature = 0.05, with TGS**

| Model | ACMU | | | | ACMU++ | | | |
|-------|------|------|------|------|------|------|------|------|
| | **P** | **R** | **F1** | **Acc** | **P** | **R** | **F1** | **Acc** |
| **C-BERT** | 0.74 | 0.73 | 0.72 | 0.80 | 0.66 | 0.66 | 0.64 | 0.75 |
| **BEAST (Ours)** | 0.89 | 0.89 | 0.88 | 0.91 | 0.86 | 0.86 | 0.84 | 0.89 |

**Table 4: Performance of C-BERT and BEAST on Drug abuse datasets: dropout = 0.1, Pooler Method = [CLS], Temperature = 0.05**

detection. The health mention datasets are categorized into three labels: 0 for Figurative, 1 for Non-Health, and 2 for Health. For binary classification, d1 retains labels 0 and 2, while d2 and d3 retain labels 1 and 2. The drug mention datasets contain four labels: 0 for Drug-abuse, 1 for Non-abuse consumption, 2 for Drug-Mention, and 3 for Figurative.

Tables 2, 3, and 4 illustrate the performance improvements of BEAST compared to the initial C-BERT method. The results indicate that, except for d3 binary, BEAST generally enhances metaphor detection across all datasets compared to the C-BERT model. Specifically, for d1 and d2 binary classifications, the F1 scores improved by 3% (from 95% to 98% for d1, and from 83% to 86% for d2), while d3 binary saw a slight decrease in F1 score from 91% (Beast) to 90% (SimCSE).

In contrast, the detection effectiveness of BEAST showed a significant improvement on the full-label HMC dataset and drug mention datasets. F1 scores for d1, d2, and d3 increased by 5% (from 89% to 94%), 13% (from 71% to 84%), and 10% (from 81% to 91%), respectively. Furthermore, the drug and new drug datasets demonstrated substantial improvements with F1 scores increasing by 16% (from 72% to 88%) and 20% (from 64% to 84%), respectively.

Furthermore, in Figures 5 and 6 we compare the clustering performance of C-BERT and BEAST. The figures reveal that while the C-BERT model is capable of clustering tweets into different labels, some tweets remain misclassified between labels. In contrast, BEAST enhances the clustering quality, effectively grouping tweets of the same type together and reducing label confusion.

## 5.1 TGS Implementation

To improve the TGS abilities of our model, we selected four datasets to self-train our model on: Wiki-Title [8], Wikipedia[2], Reddit[21], and Movie Ratings [13]. The results of this experiment are illustrated in 4 as a line chart. The graph indicates that each TGS dataset has a relatively similar impact on the model's F1 score. Except for the movie dataset, which negatively affected d2, the remaining F1 scores achieved using the TGS dataset were higher than those of the model trained without the TGS dataset. In general, the F1 scores were improved over the models that did not undergo self-training.

## 5.2 Hyperparameter Tuning

To investigate the impact of various parameters on detection performance, we examined different temperature hyperparameters, pooler methods, and dropout rates. Specifically, we tested temperature values of 0.01, 0.05, and 0.1 across the New Drug, HMC d1, d2, and d3, and d1, d2, and d3 binary datasets. We observed that a temperature of 0.1 generally achieved the highest F1 scores for all datasets except for the New Drug and d2 binary datasets.

We also explored two pooler methods: CLS, which adds a [CLS] token to the input vector to capture contextual semantic information,

---
[2]https://dumps.wikimedia.org/

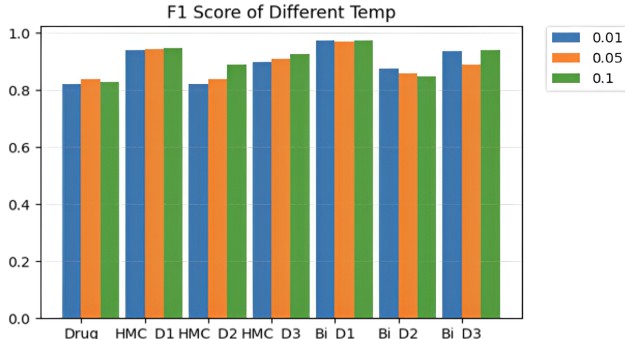

Figure 1: F1 scores with different temperature settings

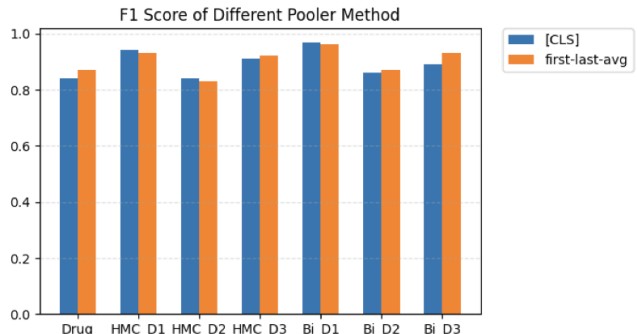

Figure 2: F1 scores with different pooler methods

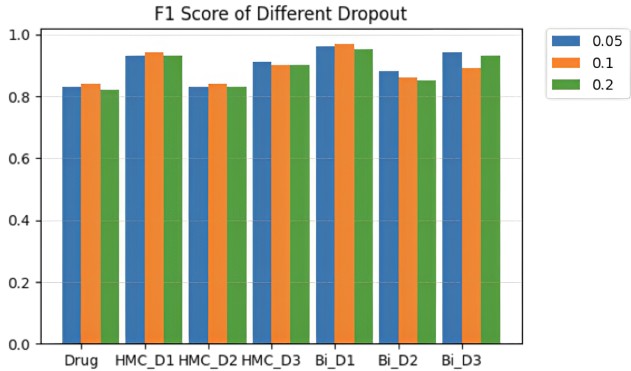

Figure 3: F1 scores with different dropouts

and First-Last-Avg, which averages the vectors from the first and last layers. Our tests revealed that First-Last-Avg performed better for the New Drug, d3, d2 binary, and d3 binary datasets, while CLS yielded higher F1 scores for the remaining datasets. Additionally, we evaluated dropout rates of 0.05, 0.1, and 0.2, finding that a dropout rate of 0.2 often resulted in the lowest F1 scores, except for the d3 binary dataset. Histograms for experiments with various temperature settings, pooler methods, and dropout settings are available in Figure 1, 2 and 3.

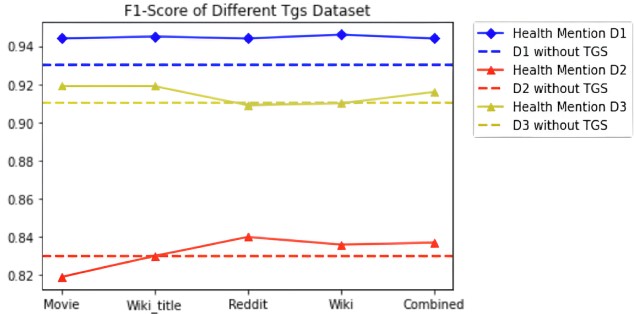

Figure 4: F1 scores of our model with different datasets for self-training.

## 5.3 Label Clusters

We visualize the representations learned by BEAST to demonstrate its effectiveness in capturing subtle semantic distinctions in the text data, highlighting the model's ability to separate figurative from literal mentions. Figure 5 illustrates the T-SNE plots for the HMC dataset, showing how BEAST creates more cohesive clusters compared to C-BERT. Figure 6 illustrates the T-SNE plots for the Drug abuse dataset, where BEAST demonstrates similarly effective clustering, clearly differentiating among the four label categories.

## 6 Conclusion

In this paper, we introduced BEAST, a model that integrates contrastive learning and unsupervised sentence embeddings for enhanced drug abuse detection on social media. By leveraging contrastive objectives alongside a Target-Based Generating Strategy, our approach outperforms a contrastive BERT baseline in identifying drug-related posts, especially those concealed through figurative language. SimCSE further refines sentence representations by constructing positive and negative pairs through subtle modifications, enabling the model to better differentiate between literal and metaphorical expressions. Experimental results on multiple datasets demonstrate the effectiveness of BEAST in consistently achieving higher performance and improved clustering, reflecting the model's deeper comprehension of nuanced language use in social media. Future work could explore domain adaptation methods to generalize BEAST to more or beyond social media platforms, as well as investigate additional data augmentation strategies for further improvement in recognizing complex linguistic phenomena such as sarcasm and code-switching. The continued evolution of contrastive learning techniques and unsupervised embeddings holds significant promise for tackling similar challenges across an array of health informatics tasks.

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

# A  Appendix

## Limitations

A key limitation of this study is the representativeness of the data, as Twitter users may not accurately reflect the broader population affected by prescription drug abuse, leading to potential biases in our findings. Additionally, the brevity and informality of social media posts can introduce ambiguity, making it challenging for machine learning models to accurately detect nuanced cases of drug abuse, especially when figurative language or sarcasm is used. Furthermore, the dataset may suffer from imbalances, with fewer posts related to actual drug misuse compared to non-abuse mentions, which could affect the model's performance and generalizability. Lastly, the reliance on publicly available data means that private conversations, which may provide more detailed insights, are not captured, limiting the depth of analysis.

## Ethical Statement

To address ethical concerns in this study, several steps were taken to ensure the protection of user privacy. While social media platforms like Twitter provide public data, users may not fully understand how their posts can be used in research. To mitigate privacy risks, we removed any user mentions or identifying information from the dataset, ensuring that personal identities remain protected. Furthermore, only tweet IDs are released, rather than the full text of tweets. This ensures that if a user deletes a tweet or their account, the content can no longer be accessed, thus respecting the user's control over their own data. Additionally, the research strictly adheres to platform-specific terms of service and ethical guidelines for using publicly available data.

Additionally, there are concerns about potential stigmatization and misuse of the research outcomes. Prescription drug abuse is a highly sensitive issue, and any misinterpretation or misclassification of users' content may contribute to harmful stereotypes or false assumptions about individuals or communities. This raises questions of fairness and bias in the models used for detecting abuse. Future researchers in this field must ensure that machine learning models are trained and tested rigorously to avoid biases that could lead to discriminatory outcomes. Furthermore, the findings of such research must be communicated responsibly, avoiding sensationalism or misrepresentation, particularly when dealing with health-related topics that could impact public perception and policy decisions.