# OpenReview forum: "BEAST: Leveraging Contrastive Learning and Unsupervised Sentence Embeddings for Improved Drug Abuse Detection"
_ACM.org/TheWebConf/2025/Workshop/TIME — TIME 2025 Oral_

### Official Review · Reviewer_b1eq · 2025-01-11
**The work introduces BEAST a novel model leveraging contrastive learning and unsupervised embeddings to enhance drug abuse detection on social media. It demonstrates high quality, clarity, originality, and significance by outperforming existing models, particularly in handling figurative language. The approach is innovative and impactful for public health, though it may face challenges with data limitations and model complexity.**

**Rating:** 7
**Confidence:** 3

**Review:**

Quality:
The work appears to be of high quality as it introduces a novel model BEAST which integrates contrastive learning and unsupervised sentence embeddings to enhance drug abuse detection on social media. The experimental results demonstrate the model's effectiveness in outperforming a contrastive BERT baseline particularly in identifying drug related posts concealed through figurative language.

Clarity:
The clarity of the work is commendable. The paper clearly outlines the challenges faced by traditional machine learning models in detecting drug abuse references on social media due to informal language, sarcasm, and figurative speech. It also explains the methodology and the improvements made by the BEAST model in a structured manner.

Originality:
The originality of the work is notable. The integration of contrastive learning with a Target-Based Generating Strategy and the use of SimCSE to refine sentence representations are innovative approaches that address the limitations of existing models in capturing nuanced language use on social media.

Significance:
The significance of this work is high, as it addresses a critical public health issue by improving the detection of drug abuse trends on social media platforms. This can lead to more effective public health interventions and monitoring.

Overall Pros:

Innovative Approach: The use of contrastive learning and unsupervised embeddings is a novel approach to tackle the challenges of detecting drug abuse references in social media.

Improved Performance: The BEAST model shows superior performance over existing models, particularly in handling figurative language.

Public Health Impact: The work has significant implications for public health monitoring and intervention.

Overall Cons:

Data Limitations: The reliance on publicly available data means that private conversations, which could provide more detailed insights, are not captured.

Generalizability: The model's applicability might be limited in scenarios where extra datasets for candidate generation are unavailable. Complexity: The integration of multiple advanced techniques may increase the complexity of the model, potentially making it more challenging to implement and understand.

Overall, the work presents a significant advancement in the field of social media analysis for drug abuse detection, with a clear and original approach that addresses existing challenges effectively

---

### Official Review · Reviewer_46HV · 2025-01-12
**Minor Revisions**

**Rating:** 6
**Confidence:** 4

**Review:**

Strength -
1. Good overall structure of manuscript
2. good result and viz shared in format for comprehensive evaluation

Weakness -
1. More mathematical and statistical evaluation would help to strengthen the paper better
2. More comparative analysis would also assist in claiming the results
3. More large scale analysis and data set could enhance multiple usecases

---

### Official Review · Reviewer_DUnU · 2025-01-13

**Rating:** 6
**Confidence:** 4

**Review:**

The overall structure of this paper is academically rigorous, well written, clearly defined problems, and clearly introduced methods. I think it can be accepted. This is more like a paper than other submissions.

---

### Official Review · Reviewer_xLTB · 2025-01-16
**Reviews from a technical rationality perspective**

**Rating:** 5
**Confidence:** 2

**Review:**

This article focuses on combining contrastive learning with drug abuse detection. From my understanding, this task is similar to a text classification task. It should be noted that I am not an expert in the field of Drug Abuse Detection; in other words, I have reviewed the article only from a technical rationality perspective, but I have not checked whether the technology presented is novel.

The strengths of this article are:

The topic is novel, addresses a sensitive issue, and is beneficial to society and public welfare.

It is well-organized and structurally complete.

Extensive comparative experiments were conducted.

The following are the weaknesses I identified in the article:

In Section 4.1, "We finetune a pre-trained instance of BERT." Does this process utilize the labels in Table 1? If so, the formula should be clearly stated.
Additionally, if the loss functions in Sections 4.2-4.3 are designed for this finetuning process, the final loss function should be clearly explained, and the first step is Eq.1.

The formula for SimCSE is not provided, and the current textual explanation does not clarify why SimCSE can address the issue of not bringing samples with the same label closer together, nor does it explain how unsupervised learning is integrated with SimCSE. I believe the corresponding formula should be added here.

If C-BERT is not used in the final model, I do not think it is necessary to dedicate a separate paragraph to it with the whole subsection. Moreover, intuitively, if the drawback of C-BERT is that it cannot bring samples with the same label closer, I believe directly combining cross-entropy loss with contrastive loss could effectively solve this issue without introducing additional techniques. The cross-entropy loss refers to the loss from the embedding to the predicted labels, and the predicted labels should be close to the true labels, which are the labels in Table 1.

Regarding the experimental section, I believe the following adjustments are needed: First, why is SimCSE not included as a baseline? Second, in some cases, your method does not seem to be optimal, which may need to be discussed in more detail. Third, Figure 5 is quite blurry and needs to be improved.

"machine learning (ML)" should be written as "Machine Learning (ML)."

---

### Official Review · Reviewer_1ZCM · 2025-01-19
**This paper introduces the BEAST model for detecting drug abuse in informal social media language, but it lacks clear distinction from existing methods, sufficient detail in the explanation, and comparisons with traditional models.**

**Rating:** 6
**Confidence:** 3

**Review:**

This article explores using social media data to monitor drug abuse, proposing a new unsupervised contrastive learning model (BEAST) to detect drug abuse hidden in informal language.

Strengths:
- Clear Problem and Motivation: The research problem is well-defined, and the paper is generally easy to follow.
- Effective Methodology: The methodology is validated through extensive experiments, demonstrating its effectiveness.

Weaknesses:
- Unclear Contributions: The paper does not clearly distinguish how it differs from existing contrastive metaphor detection methods and fails to highlight its primary contributions.
- Insufficient Detail or Errors: There is a lack of clarity regarding the objectives of TGS and self-training implementation, as well as an incorrect use of temperature coefficient symbols.
- Lack of Model Comparisons: The paper does not compare its proposed model with traditional models or other metaphor detection methods.

---

### Meta-Review · Area_Chair_J9zG · 2025-01-26

**Recommendation:** Accept (Poster)
**Confidence:** 3

**Metareview:**

This paper introduces a model called BEAST that combines contrastive learning with SimCSE to improve drug abuse detection in social media. The methodology seems to be well throughout and the experimental results seem good, showing reasonable improvements over the baseline CF models, especially detecting in figurative language but this paper heavily relies on social media datasets.

The idea of using contrastive learning with unsupervised sentenced embeddings is relatively fresh and highly relevant. The focus on natural language, metaphors or sarcasm makes this research unique. At the same time, BERT, contrastive learning and SimCSE are pretty standard, so the originality lies more in the application than the methodology.

Overall: The paper takes an important problem and does a good job. While there are some gaps in real-world applicability and ethical considerations, the work is strong enough to be shared and discussed.

Accept for Poster presentation.

---

### Decision · Program_Chairs · 2025-01-27

**Decision:**

Accept (Oral)

**Comment:**

The program chair concurs with the area chair's decision.

For the camera-ready version, please revise your paper according to the feedback provided by the reviewers.

Workshop papers must be written in English, follow a double-column format, and comply with the [ACM template](https://www2025.thewebconf.org/short-papers) and formatting guidelines. The template is also available in [Overleaf](https://www.overleaf.com/latex/templates/association-for-computing-machinery-acm-sig-proceedings-template/bmvfhcdnxfty). For authors using Microsoft Word, the Word Interim Template is recommended.

Camera-ready versions of accepted papers can and should include all information to identify authors, and should acknowledge any funding received that directly supported the presented research.

In addition, ensure that the DOI (to be provided by the PCs at a later stage) is included, and cite the workshop (to appear) using the following reference:

```
@inproceedings{time2025,
  title={TIME 2025: 1st International Workshop on Transformative Insights in Multi-faceted Evaluation},
  author={Lei Wang and Md Zakir Hossain and Syed Islam and Tom Gedeon and Sharifa Alghowinem and Isabella Yu and Serena Bono and Xuanying Zhu and Gennie Nguyen and Nur Haldar and Seyed Jalali and Abdur Razzaque and Imran Razzak and Rafiqul Islam and Shahadat Uddin and Naeem Janjua and Aneesh Krishna and Manzur Ashraf},
  booktitle={ACM Web Conference Workshop},
  year={2025}
}
```

Please note that at least one in-person registration is required for each accepted workshop paper to be included in the Companion Proceedings of WWW 2025. All accepted papers must be presented at the conference. Papers not presented (no-shows) may be withdrawn from the companion proceedings. Presentations will be conducted in two formats: oral and poster.

The camera-ready deadline for workshop papers is 7 February 2025 (AoE).